

# Positive selection within the genomes of SARS-CoV-2 and other Coronaviruses independent of impact on protein function

Alejandro Berrio[1], Valerie Gartner[1,2] and Gregory A. Wray[1,3]

[1] Department of Biology, Duke University, Durham, NC, USA
[2] University Program in Genetics and Genomics, Duke University, Durham, NC, USA
[3] Center for Genomic and Computational Biology, Duke University, Durham, NC, USA

## ABSTRACT

**Background:** The emergence of a novel coronavirus (SARS-CoV-2) associated with severe acute respiratory disease (COVID-19) has prompted efforts to understand the genetic basis for its unique characteristics and its jump from non-primate hosts to humans. Tests for positive selection can identify apparently nonrandom patterns of mutation accumulation within genomes, highlighting regions where molecular function may have changed during the origin of a species. Several recent studies of the SARS-CoV-2 genome have identified signals of conservation and positive selection within the gene encoding Spike protein based on the ratio of synonymous to nonsynonymous substitution. Such tests cannot, however, detect changes in the function of RNA molecules.

**Methods:** Here we apply a test for branch-specific oversubstitution of mutations within narrow windows of the genome without reference to the genetic code.

**Results:** We recapitulate the finding that the gene encoding Spike protein has been a target of both purifying and positive selection. In addition, we find other likely targets of positive selection within the genome of SARS-CoV-2, specifically within the genes encoding Nsp4 and Nsp16. Homology-directed modeling indicates no change in either Nsp4 or Nsp16 protein structure relative to the most recent common ancestor. These SARS-CoV-2-specific mutations may affect molecular processes mediated by the positive or negative RNA molecules, including transcription, translation, RNA stability, and evasion of the host innate immune system. Our results highlight the importance of considering mutations in viral genomes not only from the perspective of their impact on protein structure, but also how they may impact other molecular processes critical to the viral life cycle.

# INTRODUCTION

An important challenge in understanding zoonotic events is identifying the genetic changes that allow a pathogen to infect a new host. Such information can highlight molecular processes in both the pathogen and host that have practical value. The recent outbreak of SARS-CoV-2, a novel coronavirus, provides both a challenge and an opportunity to learn more about the specific adaptations that enable the virus to thrive in

Corresponding author
Alejandro Berrio,
alejo.berrio@duke.edu

human hosts and that endow it with traits distinct from previously described coronaviruses (*Andersen et al., 2020*; *Morens et al., 2020*). Formal tests for natural selection are a powerful tool in this endeavor because they can be applied in an unbiased manner throughout the viral genome: evidence of negative selection can reveal regions of the genome that are broadly constrained functionally and thus unlikely to contribute to species-specific traits, while evidence of branch-specific positive selection can identify candidate regions of the genome where molecular processes may have diverged from that of other species.

Several recent studies have tested for natural selection in the SARS-CoV-2 genome based on the ratio of synonymous to non-synonymous (dN/dS) substitutions relative to other coronaviruses (*Tang et al., 2020*; *Chaw et al., 2020*; *Li et al., 2020a*). The most prominent signal to emerge from these studies is a mix of positive and purifying selection within the gene encoding the Spike glycoprotein, which mediates invasion of host cells by binding to the angiotensin-converting enzyme 2 (ACE2) receptor in host cells (*Gallagher & Buchmeier, 2001*; *Tortorici & Veesler, 2019*). This finding makes good biological sense, because structural changes in the spike protein are common and are known to influence the ability of the virus to infect new hosts and jump between species (*Hulswit, De Haan & Bosch, 2016*). A single nucleotide polymorphism (SNP) that results in an amino acid substitution in Spike protein (A > G at 23,403 bp; D614G) has increased in frequency during the global pandemic more rapidly than other SNPs (*Korber et al., 2020*), leading to speculation that it is an adaptation that alters the interaction between Spike and ACE2, FURIN and TMPRSS2 (*Eaaswarkhanth, Al Madhoun & Al-Mulla, 2020*).

Beyond mutations that alter Spike protein, however, there exists little understanding of positive selection within the SARS-CoV-2 genome and how this may have shaped viral traits. Few convincing signals of positive selection exist for any of the other viral proteins (*Cagliani et al., 2020*; *Velazquez-Salinas et al., 2020*; *Chaw et al., 2020*). For RNA viruses, however, critical aspects of the life cycle rely on molecular processes that are not reflected in protein sequence. In particular, in positive-strand RNA viruses such as coronaviruses, the single RNA molecule that constitutes the genome is first transcribed and translated to produce the replicase polyprotein 1a and 1ab that is cleaved into multiple non-structural proteins, some of which participate in the assembly of a cellular structure known as the replicase-transcriptase complex (RTC), where the proper environment for viral replication and transcription is created. Then, the RNA-dependent-RNA-polymerase (RdRp or Nsp12) produces negative sense genomic and subgenomic RNAs that are used as template strands that are then transcribed in the opposite direction to make more positive-sense viral genomes and a variety of RNA molecules that are translated into structural proteins for packaging (*Fehr & Perlman, 2015*; *Kim et al., 2020*). Although the viral proteins that help mediate these processes are visible to tests for selection that rely on dN/dS ratios, the RNA molecules with which they interact are not. This leaves the operation of natural selection on important molecular functions within the viral life cycle largely unexamined.

In order to test for positive selection on RNA function independent of its role in coding for amino acids, we utilized a test for positive selection, *adaptiPhy* (*Berrio, Haygood & Wray, 2020*), that identifies an excess of nucleotide substitutions within a defined
window in the genome relative to neutral expectation using a likelihood ratio framework (*Wong & Nielsen, 2004*; *Haygood et al., 2007*). This test infers regions of the genome that were likely targets of branch-specific positive selection in several *Sarbecovirus* species from bat, pangolin, and human hosts. Our results recapitulate results from dN/dS-based tests that highlight *S*, the gene encoding Spike protein, as a prominent target of natural selection within the SARS-CoV-2 genome (*Cagliani et al., 2020*; *Chaw et al., 2020*; *Li et al., 2020a*). Importantly, we also identify genomic regions not previously reported to be targets of positive selection. Based on structural modeling of RNA and protein, we argue that these newly identified regions of positive selection may affect species-specific RNA, rather than protein, function. These genomic regions are candidates for understanding the molecular mechanisms that endow SARS-CoV-2 with some of its unique biological properties.

## MATERIALS AND METHODS

### Sequence alignment

To identify branch specific positive selection, it is necessary to obtain a query and a reference alignment. We downloaded six high quality reference genomes from the subgenus *Sarbecovirus* and an outgroup species (Table 1). Next, we used MAFFT (*Katoh & Standley, 2013*) plugin in Geneious Prime v.2.1 (*Kearse et al., 2012*) with default settings to build a sequence alignment. Next, we refined the alignment using a gene by gene procedure. More specifically, each coding sequence annotation (i.e., ORF1a, ORF1b, ORF3a, S, M, N, etc) is selected and realigned using the *Realign Region* tool implemented in Geneious Prime v.2.1 (*Kearse et al., 2012*) using the MAFFT (*Katoh & Standley, 2013*) option.

### Testing for positive selection

Although *adaptiPhy* was originally designed to investigate regions of complex genomes under positive selection, it can be used to identify regions of a viral sequence alignment where the foreground branch is evolving at faster rates than the expectation from the background species. We performed a selection analysis on sliding windows of 300 bp with a step of 150 bp along a sequence alignment of five reference genome sequences of coronaviruses of the subgenus *Sarbecovirus* and two sequences of Pangolin Coronavirus recently published (*Liu, Chen & Chen, 2019*; *Lam et al., 2020*). This procedure generates partitions where a tree topology can be fitted. To investigate the extent of positive selection or branches with long substitution rates along the SARS-CoV-2 genome, we used a branch-specific method known as *adaptiPhy* that was initially developed in 2007 (*Haygood et al., 2007*) and recently improved (*Berrio, Haygood & Wray, 2020*). This computational methodology makes use of a likelihood ratio test based on the maximum likelihood estimates obtained from HyPhy v2.5 (*Pond, Frost & Muse, 2005*; *Pond et al., 2020*). The branch of interest (e.g., SARS-CoV-2 branch) is used as the foreground and the rest of the alignment is used as the background. To obtain data from nucleotide substitutions alone, we used *msa_split* from PHAST (*Hubisz, Pollard & Siepel, 2011*) to remove insertions and any sequence gaps that were present in the genomes of the background virus species relative to the SARS-CoV-2 genome. The assumption for the

**Table 1 Coronavirus accessions.**

| Coronavirus species | Name used | NCBI Reference sequence |
|---|---|---|
| Severe acute respiratory syndrome coronavirus 2 isolate Wuhan-Hu-1, complete genome | SARS-CoV-2 | NC_045512.2 |
| Bat coronavirus RaTG13, complete genome | Bat-CoV-RaTG13 | MN996532.1 |
| Pangolin coronavirus isolate MP789, complete genome | Pan-CoV-GD | MT121216.1 |
| Pangolin coronavirus isolate PCoV_GX-P4L, complete genome | Pan-CoV-GX | MT040333.1 |
| *Rhinolophus affinis* coronavirus isolate LYRa11, complete genome | Bat-CoV-LYRa11 | KF569996.1 |
| SARS coronavirus Tor2, complete genome NCBI Reference | SARS-CoV | NC_004718.3 |
| Bat coronavirus BM48-31/BGR/2008, complete genome NCBI Reference | Bat-CoV-BM48 | NC_014470.1 |

**Note:**
NCBI accessions of the Coronavirus sequences used in this study

background species is the same for both the null and alternative models; specifically, only neutral evolution and negative (purifying) selection are permitted. While in the foreground, the assumptions are the same as for the background in the null model. In the alternative model, all three types of evolution are permitted (neutral evolution, negative selection, and positive selection) in the foreground of the following topology: (((((SARS_CoV_2, Bat_CoV_RaTG13), Pa_CoV_Guangdong), Pa_CoV_Guangxi_P4L), (Bat_CoV_LYRa11, SARS_CoV)), Bat_CoV_BM48). This method is highly sensitive and specific and can differentiate between positive selection and relaxation of constraint (*Berrio, Haygood & Wray, 2020*). *AdaptiPhy* requires at least three kb reference alignment for each species that is used as a putatively neutral proxy for computing substitution rates. Viruses' genomes lack non-functional regions, therefore, the most reasonable proxy for neutral evolution has to be found in the regions outside the query window. To do this, we concatenated 20 regions of 300 bp of the viral genome alignment that were drawn randomly with replacement from the entire genome alignment. Then, for each query alignment, we built a reference alignment of six kb as it produces a stable evolutionary standard of substitution rates. To control for the stochasticity of the evolutionary process, we run each query against 10 bootstrapped samples of reference alignments. Finally, we used a custom R script to compute the likelihood ratio, which was used as a test statistic for a chi-squared test with one degree of freedom to calculate a *P*-value for each query. Then, we corrected the distribution of all *P*-values per query region using the *p.adjust()* R function with the fdr method. Next, we classified a query window to be under positive selection if the *P*-adjusted value was <0.05. We were unable to successfully run *adaptiPhy* on two windows because the outgroup species (Bat_CoV_BM48) contained a deletion of 406 bp relative to SARS-CoV-2, which spans the entire ORF8.

To visualize the strength of selection comprehensively, we computed the statistic ζ (zeta), representing the evolutionary rate. To calculate this rate, we compared the substitution rate in the query with their respective reference alignments. The distribution of substitution rates for each branch and nodes in each query and reference sequence was calculated using *phyloFit* (*Hubisz, Pollard & Siepel, 2011*). Then, the ratio of substitution rate in the query is divided by the substitution rate in the reference.

This parameter, "ζ", is analogous to ω (omega), the ratio of dN/dS, where a value of ω < 1 indicates constraint or negative selection; a value of ω = 1 indicates neutrality; and a value of ω > 1 indicates positive selection (*Wong & Nielsen, 2004*).

## Testing for conservation

To test for conservation, we used the *PhastCons* computational method from PHAST (*Siepel et al., 2005*; *Hubisz, Pollard & Siepel, 2011*). To run this tool, we used the models obtained with *phyloFit* for the reference alignments and then, we generated an average estimate of the conserved and non-conserved states of the models with *phyloBoot* (*Hubisz, Pollard & Siepel, 2011*). Finally, we run the final analysis using *PhastCons* on the query alignments using the previous models to generate *PhastCons* values for each base-pair along the sequence. To plot these we took the average from each alignment and plot it using the library Gviz and Bioconductor (*Hahne & Ivanek, 2016*) in R.

## Testing for recombination

Inference of branch specific selection can be confounded by recombination given that a single phylogenetic tree may not explain the evolution of viruses. Recombination is common in coronaviruses (*Hon et al., 2008*; *Graham & Baric, 2010*; *Lau et al., 2015*; *Hu et al., 2017*; *Li et al., 2020b*; *Lam et al., 2020*) and it should be accounted for as an alternative explanation of selection at the nucleotide level. Here, we screened for evidence of recombination by estimating phylogenetic trees in sliding windows of 500 bp and a step of 150 along coronavirus alignment using RaXML-NG v0.9 (*Kozlov et al., 2019*).

## Evaluating polymorphic diversity in the pandemics of 2020

We downloaded complete sequences of SARS-CoV-2 genomes from the NCBI Virus database (https://www.ncbi.nlm.nih.gov/labs/virus/vssi/#/virus?SeqType_s=Nucleotide). As of June 26, 2020, we obtained and aligned 5,597 SARS-CoV-2 genomes sequenced worldwide. To align these sequences, we used MAFFT (*Katoh & Standley, 2013*) plugin from Geneious Prime 2.1 (*Kearse et al., 2012*), eliminating 597 sequences with the highest number of differences and ambiguities relative to the reference sequence (RefSeq: NC_045512.2), for a total of 5,000 sequences. Next, we estimated the frequency of SNP variants using the Find Variations/SNPs tool with a minimum coverage of 4,900 sequences and a minimum frequency of 0.01, to identify nucleotide variants among a subset of high quality sequenced genomes in order to evaluate ongoing evolution in the regions under positive selection.

## Analysis of RNA and protein structures

To investigate potential structural changes in Nsp4 and Nsp16 at both the RNA and protein level, we performed minimum free energy (MFE) prediction analysis using the RNAfold WebServer (*Gruber et al., 2008*; *Lorenz et al., 2011*) and consensus homology modeling using PHYRE2's intensive mode (*Kelley et al., 2016*). These analyses were performed for both Nsp4 and Nsp16 sequences for SARS-CoV-2, Bat-CoV_RaTG13, Pan-CoV-Guangdong, and SARS-CoV.

RNAfold uses a loop-based energy model and a dynamic programing algorithm to predict the structure of the sequence such that the free energy of the structure is minimized. The RNAfold WebServer generates graphical outputs for both the MFE and Centroid structures, which display the base pairing probabilities by color (blue = 0, red =1). These two MFE structures correspond to the MFE and the Centroid traces in the mountain plot, which is a positional representation of the secondary structure. In our figures, we show the MFE structure prediction.

PHYRE2 aligns input protein sequences using Position-Specific Iterated BLAST against sequences of experimentally resolved protein structures. A 3D model of the input sequence is then constructed based on homology-matched templates, optimizing for greatest sequence coverage and highest confidence. Regions of the input sequence without a matching template sequence are modeled ab initio and with Poing, a multi-template modeling tool. Pairwise comparisons of predicted protein structures were visualized using PyMOL software (*DeLano, 2002*). Alignment and structural comparisons performed by FATCAT (*Ye & Godzik, 2004*).

# RESULTS

## Positive and negative selection are highly localized within coronavirus genomes

We tested for branch-specific selection on nucleotide sequences in coronavirus genomes, focusing on six species from the *Sarbecovirus* Subgenus (Coronaviridae family) and Bat-CoV-BM48-31/BGR/2008 as an outgroup. Using 300 bp windows with a step size of 150 bp, we scanned the genome alignment for concentrations of fixed mutations that exceed the neutral expectation based on the genome as whole relative to that particular window's evolutionary history among the seven species. This test identifies regions of the genome showing the most extreme divergence in nucleotide sequence on a particular branch relative to its specific background rate of evolution across the entire phylogeny and without reference to the genetic code. Figure 1A shows windows of inferred positive selection (red dots) on the branch leading to each species. The results reveal several signals of positive selection that are unique to a single species and others that are recapitulated in multiple species. The latter finding suggests that some segments of the viral genome have repeatedly experienced adaptive modification. In general, the distribution of positive selection is more similar in closely related species than in divergent ones (Figs. 1A and 1B), suggesting that some molecular functions have been altered over an interval that extends beyond the origin of a single species but not across the entire *Sarbecovirus* radiation.

Next, we identified regions of the genome that are highly conserved across the *Sarbecovirus* genomes examined in this study using *PhastCons* (*Siepel et al., 2005*) (Fig. 2A). As with positive selection, conservation is highly localized (Figs. 2A and 2B). Based on a criterion of *PhastCons* >0.9, we found high levels of conservation in regions encoding seven proteins: 3CL-Pro, Nsp6, Nsp8, Nsp9, Nsp10, Nsp11, RdRp, ORF3a (Protein 3a), Nucleocapsid phosphoprotein (NC), and Envelope (E) (Figs. 2A–2D).

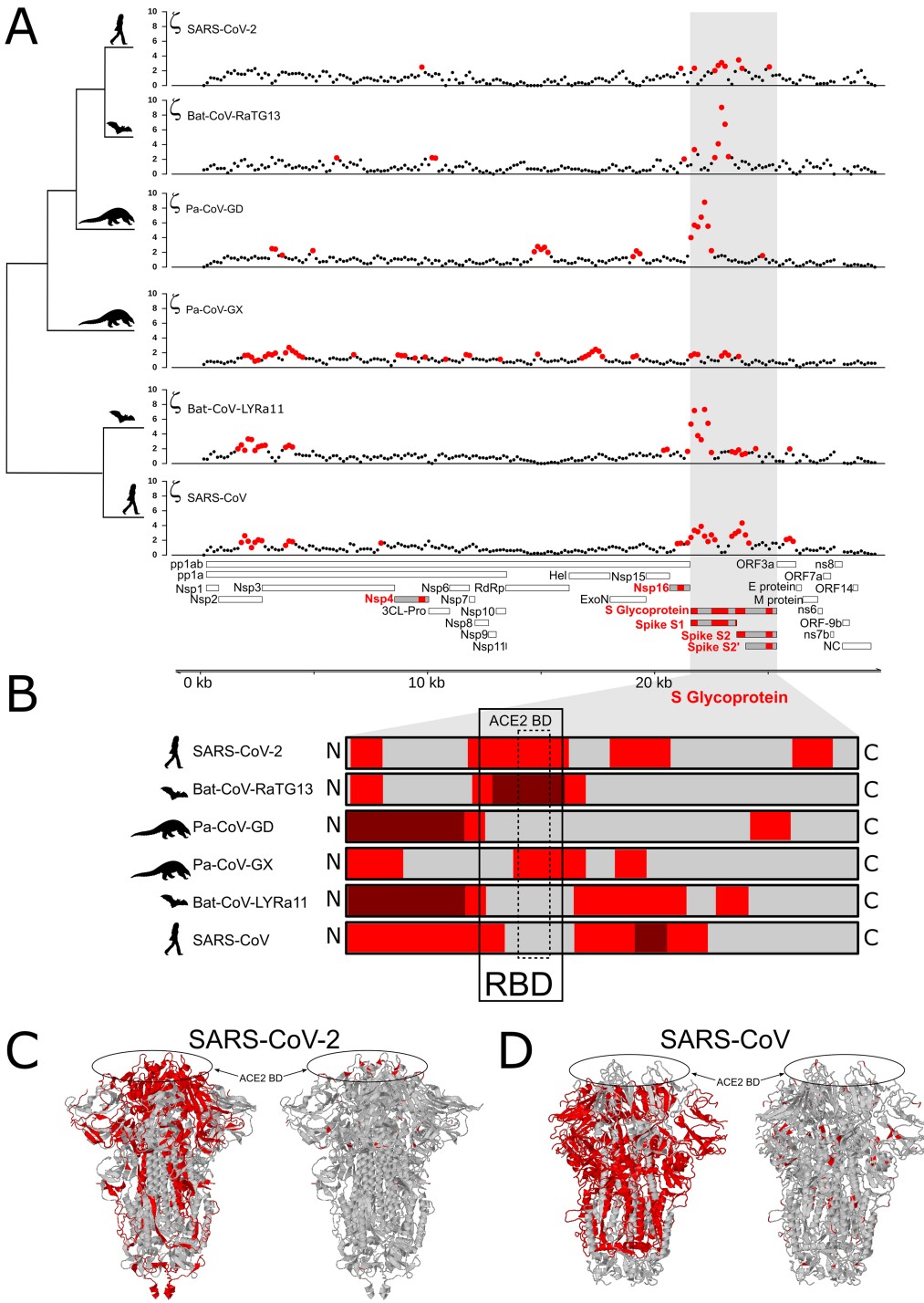

**Figure 1 Distribution of evolutionary rate and positive selection across multiple species of coronaviruses of the *Sarbecovirus* subgenus.** (A) Distribution of the evolutionary ratio, ζ, along multiple viral genome alignments. Red dots imply significant values of zeta from the *adaptiPhy* test, black dots represent neutral evolution or purifying selection in the foreground branch. (B) Visualization of selection within Spike protein among species. Dark red symbolizes windows where ζ is higher than 4, red is a significant ζ and and gray indicates neutral or purifying selection. RBD, receptor binding domain. Tertiary structure of Spike protein depicting the location positive selection and amino acid substitutions in (C) SARS-CoV-2 and (D) SARS-CoV.
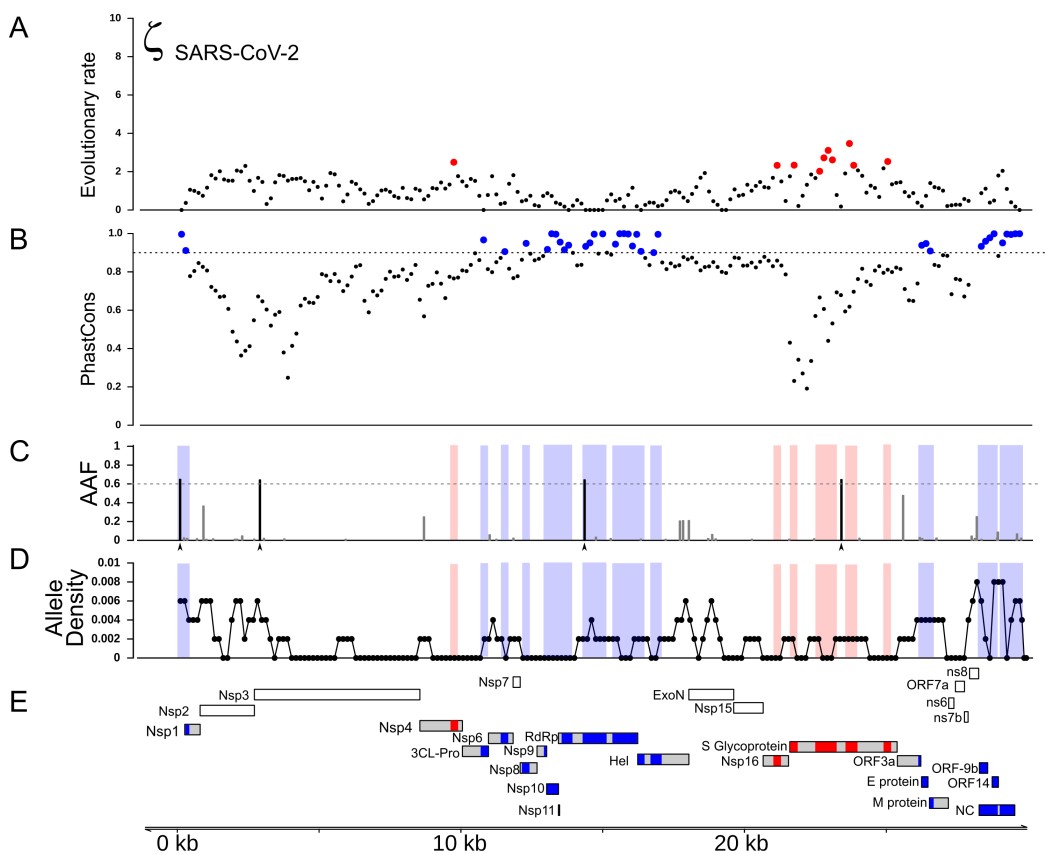

**Figure 2 Distribution of positive selection, *PhastCons* conservation, and polymorphic variation across the SARS-CoV-2 genome.** (A) Evolutionary rate ($\zeta$) with sites under significant branch specific selection as red dots. (B) Panel depicting conservation values (*PhastCons*) with highly conserved windows (*PhastCons* >0.9) as blue dots over the dashed line along the SARS-CoV-2 genome. (C) Alternative allele frequency for 5,000 high quality genomes available in NCBI. Dotted line represents an arbitrary threshold of 0.6 and SNPs in strong linkage disequilibrium are highlighted with arrowheads under black bars. (D) Allele density in windows of 500 bp with a step of 150 bp. Red boxes in B and C symbolize regions under positive selection, while blue boxes represent high conservation. (E) Annotations for all the mature proteins known to be expressed in SARS-CoV-2.

These loci of exceptional sequence conservation highlight critical molecular features: NC and E are essential structural proteins of the coronavirus capsid, while the other proteins regulate a variety of molecular process during viral replication (*Tan et al., 2005*; *Lu et al., 2006*; *Minakshi et al., 2009*; *Freundt et al., 2010*; *Fuchs, 2012*; *Yue et al., 2018*).

Because new mutations emerge and new strains replace older ones, we next investigated how much the specific strain used to represent SARS-CoV-2 influences test results. We re-ran the tests for positive selection using a strain of SARS-CoV-2 that contains four derived SNPs that commonly co-occur in currently circulating strains. Using this strain did not change the distribution of inferred regions of positive selection during the origin of SARS-CoV-2 (Fig. S1). We also generated two artificial genomes where we added four and nine mutations in the vicinity of site 14,408 to test the sensitivity of the test. We found that as zeta increased within the window, the test turned significant when more than five mutations are added (Fig. S1). It is important to note that the exact number of

mutations that produce a significant test result may differ in other regions of the genome, depending on the degree of sequence conservation among species.

## The gene encoding Spike protein is under persistent positive selection

In all ingroup species examined we detected signals of positive selection within the S gene, which encodes the Spike protein. With the exception of Pa-CoV-GX, this was the most prominent signal in the entire genome (Fig. 1A). This finding confirms previous studies that used the dN/dS ratio to test for selection on protein function (*Tang et al., 2020*; *Chaw et al., 2020*; *Li et al., 2020a*). Interestingly, we observed that the specific regions showing signatures of positive selection differed between species (Figs. 1B and 1C). In SARS-CoV-2, we detected signals of positive selection in four segments of the S gene. First, the region encoding the entire receptor binding domain (RBD) shows an extended signal (Figs. 1B, 1C and 3A); as others have noted, structural changes in this region may improve binding to human ACE2 (*Wang et al., 2020*; *Wrapp et al., 2020*; *Wang, Liu & Gao, 2020*). The second segment encodes another externally facing region, the S1 subunit N-terminal (NTD) domain, which includes the first disulfide bond (amino acids 13–113) and several glycosylation sites. The third signal of positive selection within S is located around the derived furin cleavage site (amino acids 664–812) that has been found to be essential for infection of lung cells (*Hoffmann, Kleine-Weber & Pöhlmann, 2020*). The fourth signal is located in a segment encoding the S2 and S2' subunits that includes the Heptad repeat 2 (amino acids 1114–1213). These heptad repeats were previously associated with episodes of selection for amino acids that increase the stability of the six-helix bundle formed by both heptad repeats in MERS and other coronaviruses (*Forni et al., 2015*); they are also thought to determine host expansions and therefore, facilitate virus cross-species transmission (*Graham & Baric, 2010*).

The distribution of inferred positive selection in the S gene of SARS-CoV differed from that of SARS-CoV-2 described above. Notably, there was no signal in the ACE2 binding domain (Figs. 1B and 1C). Moreover, a signal was present throughout the N-terminal domain and in the boundary region between the S1 and the S2 subunits (Fig. 1), a region that includes the proteolytic cleavage (*De Haan et al., 2004*). Interestingly, this region evolved a novel furin cleavage site in SARS-CoV-2 that may increase the cleavage efficiency and cell-cell fusion activity and changes in the virulence of the virion as seen in mutant studies of SARS-CoV and SARS-CoV-2 (*Follis, York & Nunberg, 2006*; *Hoffmann, Kleine-Weber & Pöhlmann, 2020*).

## Genes encoding Nsp4 and Nsp16 contain branch-specific signals of positive selection

We also detected two shorter signals of positive selection within the SARS-CoV-2 genome that are located outside of the S gene, in pp1ab and pp1a (Fig. 1A). Interestingly, both encode small proteins that contribute to viral replication. The first is Nsp4, which encodes a membrane-bound protein with a cytoplasmic C-terminal domain; it is thought to anchor the Viral-Replication-Transcription Complex (RTC) to the modified endoplasmic

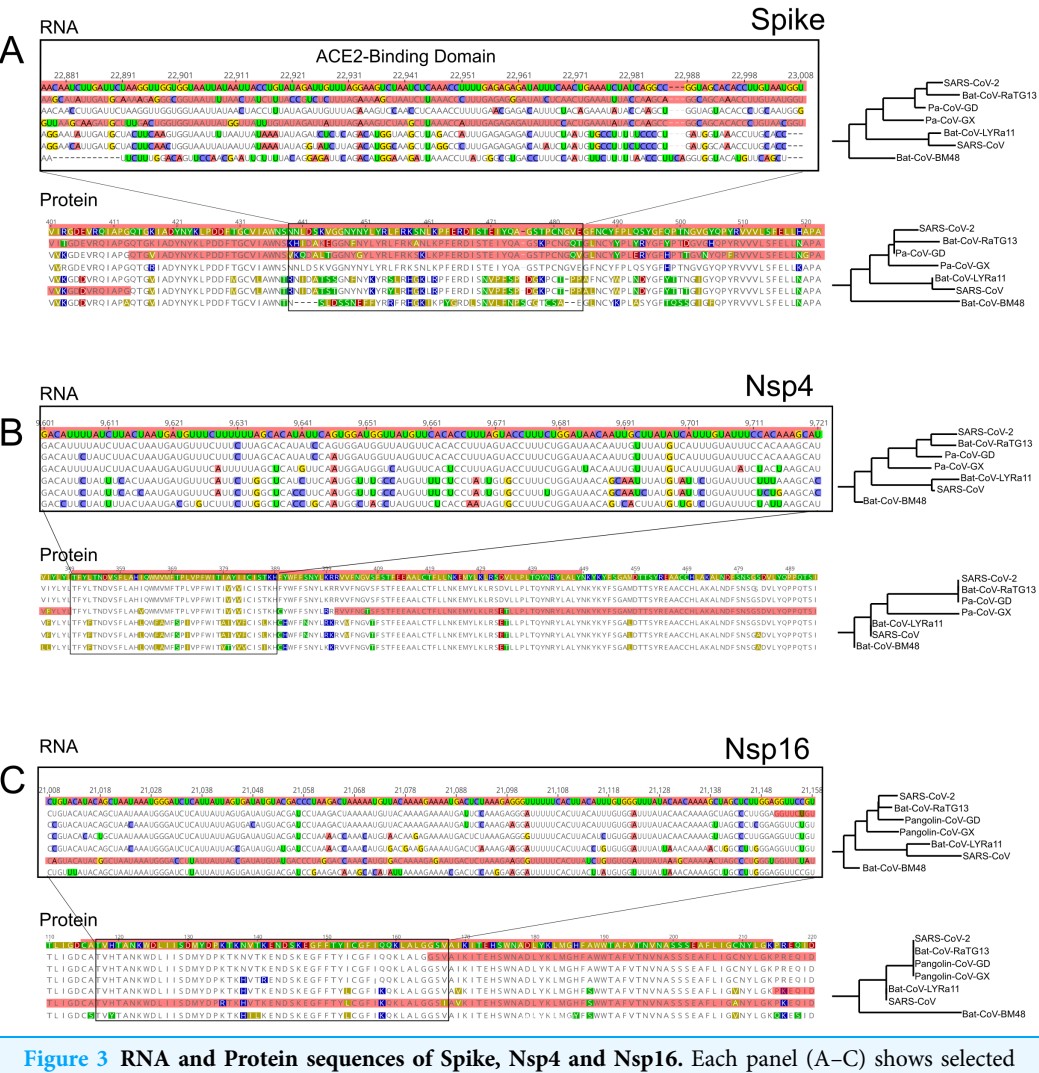

**Figure 3 RNA and Protein sequences of Spike, Nsp4 and Nsp16.** Each panel (A–C) shows selected RNA and protein sequences scoring high for positive selection in the SARS-CoV-2 branch and other branches (highlighted in red). Changes with respect to SARS-CoV-2 are highlighted in different colors.

reticulum membranes in the host cell (*Oostra et al., 2008*; *Hagemeijer et al., 2011*, *2014*; *Snijder, Decroly & Ziebuhr, 2016*). The SARS-CoV-2 Nsp4 protein differs from that of closely related sarbecoviruses by two nearly adjacent amino acids: V380A and V382I. Although this region of the genome as a whole is not highly conserved (Fig. 2), both of these positions are V residues in all of the in-group species we examined except SARS-CoV-2 (Fig. 3B; Fig. S2A). This signal is too weak to be scored as positive selection using dN/dS-based tests (*Tang et al., 2020*; *Chaw et al., 2020*; *Li et al., 2020a*) and indeed may not affect protein function given the biochemically similar side-chains of the amino acids involved.

The second signal of positive selection outside of the S gene lies within Nsp16. This gene encodes a 2′-O-methyltransferase that modifies the 5′-cap of viral mRNAs (*Decroly et al., 2008*; *Bouvet et al., 2010*) and assists in evasion of the innate immune system of host

cells (*Züst et al., 2011*; *Menachery, Debbink & Baric, 2014*; *Nelemans & Kikkert, 2019*). Of note, this is the only signal of positive selection within the SARS-CoV-2 genome that lacks any nonsynonymous substitutions (Fig. 3C), and thus could not have been detected by any test that relies on the dN/dS ratio. All of the nucleotide substitutions in Nsp16 during the origin of SARS-CoV-2 are synonymous, while the Nsp16 genes of SARS-CoV-2, Bat-Cov-RaTG13, and Pan-CoV-GD (Guangdong) all encode identical proteins (Fig. 3C; Fig. S3A). This suggests a complex mechanism of selection in the form of purifying selection at the protein level and branch-specific positive selection at the nucleotide level. Ancestral state reconstruction of Nsp16 indicates that 20 synonymous substitutions likely occurred in the lineage leading to SARS-CoV-2 after the split from the common ancestor with BatCoV-RaTG13, while 19 substitutions are synonymous substitutions that occurred in the lineage leading to Bat-CoV-RaTG13 (Supplemental Data). Eleven of these twenty substitutions are concentrated within the region scoring high for positive selection in SARS-CoV-2 and twelve within the positively selected region in Bat-CoV-RaTG13.

As a consequence, we hypothesized that the Nsp16 RNA secondary structure may differ among species in ways that affect molecular functions mediated directly (although not solely) by RNA, such as replication, transcription, translation, or evasion of the host immune system. To investigate this possibility, we first compared the secondary structure and minimum free energy (MFE) of RNA in the vicinity of Nsp4 and Nsp16 among the genomes of SARS-CoV-2, Bat-CoV-RaTG13, Pan-CoV-GD, and SARS-CoV using RNAfold (*Gruber et al., 2008*). Both the predicted secondary structures and mountain plots, which show the free energy predictions along the length of the sequence by position, reveal differences in RNA folding dynamics across the four species (Figs. S2B and S3B). Analysis of the reconstructed sequence of the SARS-CoV-2 + Bat-CoV-RaTG13 ancestor reveal that most of these differences evolved during the origin of SARS-CoV-2 (Fig. S4). These differences among species in predicted secondary structures within Nsp4 and Nsp16 stand in contrast to the 5′ UTR, which is thought to fold into a stable secondary structure that is markedly conserved among *Sarbecovirus* species (Fig. S5). Though the accuracy of MFE predictions is too low to conclusively determine whether there are real between-species differences in the RNA structures of these loci (*Mathews, 2005*), these observations suggest that the signal of positive selection within Nsp16 in the SARS-CoV-2 genome may reflect changes in RNA, rather than protein, function that are unique to this species of coronavirus.

While the focus here is on SARS-CoV-2, it is worth noting that we also detected signals of positive selection outside of the S gene in the other *Sarbecovirus* genomes examined here. The distribution of positive selection in the genome of SARS-CoV, for instance, shows some similarities to, but also notable differences from, that of SARS-CoV-2 (Fig. 1). In both species, S and Nsp16 contain signals of positive selection, although in distinct regions of the two genes (Fig. 1). In addition, the genome of SARS-CoV contains signals of positive selection in Nsp2, Nsp3, and ORF3a, none of which shows elevated rates of substitution in SARS-CoV-2. The first two genes encode proteins with important roles in viral replication: Nsp2 may disrupt intracellular signaling in the host cell (*Cornillez-Ty*
*et al., 2009*) while Nsp3 cleaves itself, Nsp1, and Nsp2 from the replicase polyproteins (*Báez-Santos, St. John & Mesecar, 2015*), assists in the assembly of the double membrane vesicles of the RTC system (*Hagemeijer et al., 2014*), and antagonizes the host innate immune response (*Tsuchida, Kawai & Akira, 2009*; *Frieman et al., 2009*; *Matthews et al., 2014*).

## Recombination does not account for most signals of positive selection

Recombination from another species can be a confounding factor in the inference of positive selection using the framework employed here, because the inserted genomic segment may be more divergent than the rest of the foreground genome is from nearby background species. Several instances of recombination have been reported in coronaviruses, including SARS-CoV-2 (*Hon et al., 2008*; *Lam et al., 2020*; *Boni et al., 2020*; *Li et al., 2020a*), making it important to distinguish regions of recombination from positive selection. The two processes produce distinct genetic signatures, with recombination the result of a single event (possibly later further recombined) and positive selection as detected here the result of multiple independent mutations that were fixed over an extended interval and are spatially concentrated. In order to test for regions of the SARS-CoV-2 that contain recombined segments from other species, we estimated the phylogenetic history of 500 bp segments of the genome with a step size of 150 bp among the aligned genomes of the seven species examined in this study. We used RaXML-NG v0.9 (*Kozlov et al., 2019*) to reconstruct topology for each segment independently and searched for cases where the topology differed from the expected topology based on the entire genome: (Bat-CoV-BM48, (Bat-CoV-LYRa11, SARS-CoV), (Pa-CoV-GX, (Pa-CoV-GD, (SARS-CoV-2, Ba-CoV-RaTG13)))). Recombination from a divergent species should produce an incongruent topology in one or more adjacent windows, revealing a recombined region and its approximate breakpoints. We identified 12 regions where the topology differed from the expected (Fig. 4). Of note, these regions are somewhat more concentrated in the part of the genome that encodes structural proteins. Consistent with a previous report (*Li et al., 2020a*), we observed overlap between regions scoring high for positive selection and recombination in S, the gene encoding Spike protein (Fig. 4M), specifically the region that encodes for the ACE2 binding domain and a region that includes the furin-cleavage site (Figs. 4F and 4G). Importantly, however, none of the putatively recombined regions overlap with the windows scoring high for positive selection within the genes encoding Nsp4 and Nsp16 proteins in SARS-CoV-2.

## Recent changes in allele frequency may result from positive selection and hitch-hiking

To gain insight into the evolutionary mechanisms that have shaped genetic variation more recently within the SARS-CoV-2 genome, we compiled a list of known mutations, based on 5,000 accessions sequenced since the beginning of the current pandemic (see Methods). As expected, the vast majority of variants are singletons, representing either mutations that are not segregating or sequencing errors. The density distribution of polymorphisms

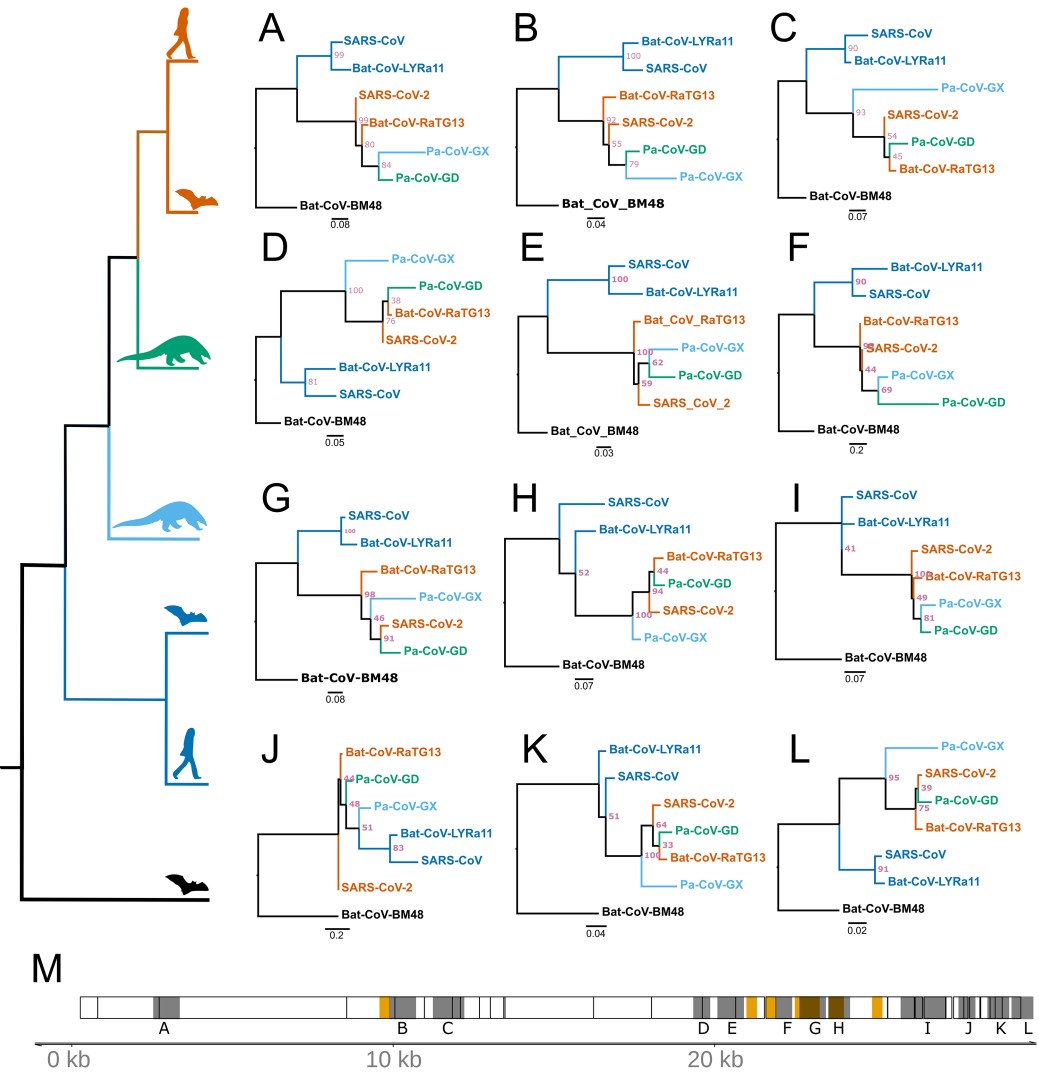

**Figure 4 Regions of coronavirus genomes that violate the species tree.** The species tree topology is shown on the left. (A–L) Tree topologies that were different from the expected topology. (M) Coronavirus genome track where the regions scoring high for positive selection in SARS-CoV-2 are highlighted in orange, regions with unexpected tree topologies highlighted in dark gray.

(regardless of frequency) is elevated within 2–3 kb at both ends the SARS-CoV-2 genome (Fig. 2D) and the site-frequency spectrum is strongly left-skewed (Fig. S6).

We next investigated the likely consequences for altered molecular function due to each of these four high-frequency derived SNPs. Two are located within regions of the genome that are highly conserved among *Sarbecovirus* species (Figs. 2B and 2C). The first is a C > U substitution at position 241 in the 5′UTR, a region of the genome where RNA secondary structure is highly conserved across Coronavirus species (*Madhugiri et al., 2016*; *Rangan et al., 2020*; *Alhatlani, 2020*). Using RNAfold (*Gruber et al., 2008*) we found that this C > U transition had no impact on the stem-loop structure established for SARS-CoV (Fig. S5). The other mutation in a conserved region of the genome is a nonsynonymous

substitution in the RdRp gene (14,408; P323L) at the interface domain, which is though to mediate protein-protein interactions (*Pachetti et al., 2020*; *Hillen et al., 2020*). Because proline residues can influence secondary structure, we used PHYRE2 to predict the impact of the P232L mutation on protein structure. Comparison of the two predicted structures using FATCAT shows they are nearly identical (Table S1). The other two high-frequency derived SNPs are located in regions that are neither highly conserved nor highly divergent. One is a synonymous SNP in Nsp3 (3,037) and the other a nonsynonymous SNP in S (23,403; D614G). This last SNP effectively removes a charged side-chain between the receptor binding domain and the furin cleavage site of S, a region of recurrent positive selection among the *Sarbecovirus* species we examined. Thus, of the four high-frequency derived SNPs, the nonsynonymous substitution in S the most plausible candidate for altering molecular function and thus becoming a target of natural selection.

## DISCUSSION

A crucial feature contributing to the global spread of COVID-19 is that viral shedding starts before the onset of symptoms (*He et al., 2020*); in contrast, shedding began 2–10 days after the onset of symptoms during the SARS epidemic of 2003 (*Peiris et al., 2003*; *Pitzer, Leung & Lipsitch, 2007*). This striking difference suggests that one or more molecular mechanisms during host cell invasion, virus replication, or immune avoidance may have changed during the origin of SARS-CoV-2. Mutations contributing to viral transmission would likely be favored by natural selection, making tests for positive selection a useful tool for identifying candidate genetic changes responsible for the unique properties of SARS-CoV-2. Here, we searched for regions of possible positive selection within the genomes of six coronavirus species, including SARS-CoV and SARS-CoV-2. The method we used tests for an excess of branch-specific nucleotide substitutions within a defined window relative to a neutral expectation for divergence in that window and without regard to the genetic code (*Wong & Nielsen, 2004*; *Haygood et al., 2007*; *Berrio, Haygood & Wray, 2020*).

Several prior studies have identified *S*, the gene encoding the Spike glycoprotein, as a target of recurrent positive selection in coronavirus genomes, including SARS-CoV-2, based on ω, the ratio of synonymous to nonsynonymous substitutions (*Andersen et al., 2020*; *Cagliani et al., 2020*; *Tang et al., 2020*; *Armijos-Jaramillo et al., 2020*; *Li et al., 2020a*). S thus serves as a positive control for our ability to detect signals of positive selection using a different approach, which considers mutations without respect to the genetic code and uses a likelihood ratio framework to identify regions of elevated, branch-specific nucleotide substitution rates relative to a model that allows only drift (*Wong & Nielsen, 2004*; *Haygood et al., 2007*; *Berrio, Haygood & Wray, 2020*). Consistent with this expectation, we found that portions of the gene encoding Spike showed a striking elevation of sequence divergence relative to the rest of the genome on the branches leading to all six species examined. The specific regions of S containing high divergence differs markedly, however, among species (Fig. 1B). In SARS-CoV and Bat-CoV-LYRa11, these
regions include the N-terminal region, which contains glycosylation sites important for viral camouflage (*Watanabe et al., 2019*; *Yang et al., 2020*) and a site of proteolytic cleavage that allows entry into the host cell (*Belouzard, Chu & Whittaker, 2009*) (Figs. 1C and 3A). In contrast, signals of positive selection in SARS-CoV-2 and Bat-CoV-RaTG13 are concentrated in the domain that mediates binding to the host receptor ACE2 (Figs. 1C and 3A). These distinct distributions suggest that modifications in different aspects of Spike function took place as various coronaviruses adapted to novel hosts. In particular, the concentration of derived amino acid substitutions in the receptor binding domain of Spike (Figs. 1B and 1C) in SARS-CoV-2 and Bat-CoV-RaTG13 may reflect selection for amino acid substitutions that result in higher affinity for ACE2 protein in different host species.

Importantly, we also detected signals of positive selection in two additional regions of the SARS-CoV-2 genome, specifically within the genes encoding Nsp4 and Nsp16 (Figs. 1A and 2A). Of note, the Nsp16 region also shows a parallel signal of positive selection on the branch leading to SARS-CoV. To our knowledge, this is the first report of possible adaptive change in molecular function during the evolutionary origin of SARS-CoV-2 outside of the gene encoding Spike protein. Prior scans for positive selection within the SARS-CoV-2 genome used elevated ω as the signal of positive selection, which restricts attention to positive selection based on changes in protein function. For coronaviruses this is a notable limitation, given that many aspects of the lifecycle involve RNA function (*Madhugiri et al., 2016*; *Ziv et al., 2020*; *Alhatlani, 2020*). In addition, the secondary structure of some segments within the RNA genome is well conserved among coronavirus species, which implies a functional role (*Rangan et al., 2020*; *Sanders et al., 2020*; *Huston et al., 2020*). Indeed, the SARS-CoV-2 genome is reported to contain more well-structured regions than any other known virus, including both coding and noncoding regions of the genome (*Huston et al., 2020*). We therefore examined nucleotide substitutions within regions of putative positive selection in Nsp4 and Nsp16 for their likely impact on both protein and RNA structure (Figs. S2 and S3).

In the case of Nsp4 protein, two nearly adjacent nonsynonymous substitutions at residues 380 and 382 occurred on the branch leading to SARS-CoV-2 (Fig. 3B). These both involve changing side chains with similar biochemical properties, respectively valine to alanine and valine to isoleucine. Homology-directed modeling of protein structure suggests that these two amino acid substitutions have very little impact on either secondary or tertiary structure when comparing the SARS-CoV-2 protein orthologue to those of the other species examined (Fig. S2A). In the case of Nsp16 protein, no nonsynonymous substitutions evolved on the branch leading to SARS-CoV-2. Thus, the signal of positive selection within Nsp4 is unlikely to reflect changes in protein structure or function, while the signal within Nsp16 cannot affect either because the encoded polypeptide is identical (Fig. 3C; Fig. S3A).

With highly similar and identical protein structures predicted for Nsp4 and Nsp16, respectively, we considered the possibility that the signals of positive selection instead reflect changes in RNA structure and function. Previous studies found that neither the

Nsp4 nor Nsp16 regions stand out as particularly well folded regions of the genome, although Nsp16 does contain a single well-folded region and Nsp4 two moderately well folded regions (*Rangan et al., 2020*; *Huston et al., 2020*). Further, both genes show significantly decreased sequence divergence among coronavirus species within predicted double-stranded region (*Rangan et al., 2020*; *Sanders et al., 2020*; *Huston et al., 2020*). Indeed, the well-folded region within Nsp16 is the only such region in the SARS-CoV-2 genome that is also well conserved among related coronaviruses (*Sanders et al., 2020*). These published observations suggest possible functional roles for folded structures within Nsp4 and Nsp16. While we have not taken a robust experimental approach to determine between-species differences in RNA secondary structure, our in silico minimum free energy (MFE) predictions suggest that the likely secondary structure of the RNA genome in the region of the Nsp4 and Nsp16 genes may differ among the six coronavirus species we examined (Figs. S2B and S3B, top rows). The MFE predictions also indicate differences among species in entropy across the regions containing the signals of positive selection, indicating possible differences in the stability of the folded molecule (Figs. S2B and S3B, bottom rows). Together, these results indicate that the folded regions of Nsp4 and Nsp16 in the SARS-Cov-2 genome may differ in shape from those of related coronaviruses.

Unfortunately, little is currently known about the molecular functions of secondary structures in coronavirus genomes. Most of the attention has been directed towards the 5′ UTR, 3′ UTR, and frameshift element at the junction between ORF1a and ORF1b, which together contain the most well-folded regions in the SARS-CoV-2 genome (*Andrews et al., 2020*; *Sanders et al., 2020*; *Huston et al., 2020*). Thus, it is not possible at this time to link structural and thermodynamic features within Nsp4 and Nsp16 that are unique to SARS-CoV-2 to specific molecular functions. As discussed above, however, published evidence suggests that RNA secondary structures within these regions of the genome may be functional (*Rangan et al., 2020*; *Sanders et al., 2020*; *Huston et al., 2020*). These functions could, in principle, affect genome or transcript function, or both. Plausible possibilities include secondary structures that recruit specific RNA-binding proteins to mediate transcriptional regulation or transcript processing (*Pirakitikulr et al., 2016*; *Pan et al., 2020*), that mediate looping for other reasons (*Gebhard, Filomatori & Gamarnik, 2011*; *Ziv et al., 2020*), or that simply facilitate or impede processivity of the replication or translation machinery (*MacFadden et al., 2018*).

To investigate what evolutionary mechanisms are shaping the genetic variation at the population level, we examined known mutations among 5,000 accessions from NCBI. Given that the effective population size of SARS-CoV-2 is likely very large, the alternative allele distribution (Fig. S6) suggests that most SNPs are not subject to positive selection and that negative selection prevents most new mutations from rising in frequency due to drift, although this may change as additional whole genomes are examined. However, we did observe four SNPs that are present at high alternative allele frequency (AAF > 0.6) (Fig. 2C), a situation that can reflect positive selection, drift, or hitch-hiking. Interestingly, all four of these SNPs are in tight LD (*Toyoshima et al., 2020*), which suggests that positive selection on one of them may have driven the other three to high

frequency due to hitch-hiking. Based on molecular modeling, the high-frequency derived mutation in S is the most plausible to be under positive selection, while the other three may be elevated due to hitch-hiking.

## CONCLUSIONS

Scans for positive selection typically focus on changes in protein function and far less often consider the possibility of adaptive change in RNA function. By shining a light on regions of the SARS-CoV-2 genome that appear to be under positive selection yet are unlikely to alter protein function, our results illustrate the value of evaluating the potential for adaptive changes in secondary structures within the genomes of RNA viruses. In particular, we identify Nsp4 and Nsp16 as regions of the SARS-CoV-2 genome that may contain mutations that contribute to the unique biological and epidemiological features of this recently emerged pathogen.

While it is tempting to speculate about the possible adaptive role of changes in RNA structure within these accelerated regions, we suggest that this is best done in the context of relevant experimental results. For example, it might be informative to modify the primary sequence of the genome so as to encode the same protein sequence while altering or disrupting secondary structure within Nsp4 or Nsp16, then assay the consequences for viral replication and for specific molecular functions. We hope that our results inspire these or other experiments aimed at better understand the evolving functions of RNA secondary structure within the SARS-CoV-2 genome.

## ACKNOWLEDGEMENTS

The authors would like to thank all the members of the lab groups of David McClay and Greg Wray for comments.

### Funding

This work did not have sources of direct funding but publishing costs were supported by the COPE fund at Duke University. The funders had no role in study design, data collection and analysis, decision to publish, or preparation of the manuscript.

### Grant Disclosures

The following grant information was disclosed by the authors:
Duke University.

### Competing Interests

The authors declare that they have no competing interests.

### Author Contributions

- Alejandro Berrio conceived and designed the experiments, performed the experiments, analyzed the data, prepared figures and/or tables, authored or reviewed drafts of the paper, and approved the final draft.

- Valerie Gartner conceived and designed the experiments, performed the experiments, analyzed the data, prepared figures and/or tables, authored or reviewed drafts of the paper, and approved the final draft.
- Gregory A. Wray conceived and designed the experiments, analyzed the data, authored or reviewed drafts of the paper, and approved the final draft.

## Data Availability

The raw data are available in the Supplemental Files.

The analytical pipelines and raw data are available in GitHub: https://github.com/wodanaz/adaptiPhy/blob/master/applications/.

## Supplemental Information

Supplemental information for this article can be found online at http://dx.doi.org/10.7717/peerj.10234#supplemental-information.

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
