# Peer review of "Positive selection within the genomes of SARS-CoV-2 and other Coronaviruses independent of impact on protein function"

_PeerJ, doi:10.7717/peerj.10234_

## Round 0.1 · original submission · Minor Revisions

Two reviewers suggest minor revision mainly on text presentation. I worry about the limited data used for analysis. New full genome sequences for SARS-CoV-2 coming recently increasing the statistical base for analysis. Please during revision ensure the result on selection remains the same taking into account new sequences or tone down your conclusion.

·

Basic reporting

Overall the manuscript is very well written, clear and easy to understand. The manuscript, when published, will definitely greatly aid in the ongoing research pertaining to the evolution and adaptability of SARS-CoV-2. However, I felt that the authors could have included more references in the manuscript, especially in the Introduction section. Also, some sentences are long and could have been broken into two to improve understanding.

Experimental design

The method section is very well written with sufficient details.

One query that I have is branch-specific dN/dS implemented in PAML package can also perform selection tests. It is unclear to me why the authors did not use this popular package. The authors should clearly mention the technical and biological superiority of their method over more commonly used Branch-specific dN/dS approach implemented in PAML.

Validity of the findings

The data are robust and statistically sound. All conclusions are well stated and linked to the original research questions.

Additional comments

1. The first paragraph of the Introduction section should have a reference.
2. Lines 63-64: "A single nucleotide polymorphism (SNP)..." - Mention the nucleotide and amino acid substitution related to this SNP
3. Line 85: "Here we utilize a test for positive selection..." - Is there a name for this test? If so, can you mention it here and describe why it is superior to other similar tests?
4. Line 104: Please elaborate the gene by gene alignment procedure
5. Lines 147-152: Can you compare and contrast Zeta obtained from PhyloFit with Omega, generated in brach-specific dN/dS implemented in PAML?
6. Line 223: I could not find Protein 3a annotated in Fig. 2. Is it annotated in a different name?
7. Lines 230-233: Fig. 1A- For human SARS-CoV2, the signal of positive selection is not as prominent as in bats and pangolin. I would call it an intermediate signal between Pa-CoV-Gx and human SARS-CoV. Then how can you call it a 'prominent' signal?
8. From Fig. 1B it seems like the fourth signal near C-terminal domain is unique to SARS-CoV2. Is that true? If so, can you discuss the potential reason(s)?
9. Please annotate the ACE2 binding domain in Fig. 1B and IC
10. Please annotate and demarcate ORF1a and ORF1b clearly in Fig. 1A
11. Lines 275-277: "Of note, this is the only...non-synonymous substitutions" - It is unclear to me if there is no non-synonymous substitution, then how did you conclude it to be a signal of positive selection? Please explain.
12. Lines 279-281: "This suggests a complex...positive selection at the nucleotide level" - How is this possible? Please explain this.
13. Please avoid using adjectives such as 'Surprised'.
14. Line 352: "However, we did observe four SNPs..." - Please highlight these SNPs in Fig. 2C. By eyeballing, at least two of these SNPs does not seem to have AAF>0.6.
15. Also, two of these SNPs lie within areas with signals of strong purifying selection. It is unclear to me how hitchhiking is possible for these SNPs. Hitchhiking, by definition can operate on neutral sites or sites with slight negative selection. Please explain with references.

Reviewer 2 ·

Basic reporting

The manuscript is excellently written - really well done.

Experimental design

One small question on the stability of predictions - more below in the General Comments.

Validity of the findings

One concern about the validity of the RNAfold predictions - more information in the General Comments

Additional comments

This manuscript reports the use of a new analysis method to identify selection pressures on the evolution of coronavirus genomes that recapitulates and enriches-upon/complements other results from protein-centered approaches. This is a well-crafted study (and extremely well-written manuscript) that provides interesting results that focus on evolutionary pressures that shape the RNA molecules comprising the viral genomic material. By focusing on features other than protein coding potential, the authors unlock insights into other aspects of coronavirus biology: e.g. functional RNA secondary structure. The results, particularly the identification and clustering of sites of positive and purifying selection, are of immense importance to the community and would make this a valuable contribution when published. The implications of RNA structure are absolutely worth discussing, however, the analyses of secondary structure presented here are (I believe) too speculative/preliminary for inclusion as a published Result.

Major Points:

1) As a non-specialist, it seems surprising that significant evolutionary pressures can be identified with an analysis of an alignment with so few genome sequences. Apologies if I missed this in the manuscript, but, it would be good to have an idea of how robust the identified sites are. For example, if you select different sequences from the same species (with some internal variation) to include in the analysis, would the results be the same: i.e. how volatile would the results be for resampled sequences?

2) There are two analyses of conserved/functional RNA structural motifs available for SARS-CoV-2: described in the Rangan et al paper, and in the preprint from Andrews et al, both cited in the manuscript. The sites of unusual selection identified in SARS-CoV-2 that were identified could be cross-referenced to structural motifs identified in these studies and that are available from the Das or Moss labs, respectively. For the Andrews, et al dataset, for example, selection sites can be loaded as a track on the RNAStructuromeDB SARS-CoV-2 genome (https://structurome.bb.iastate.edu/sars-cov-2) to aid in the analysis.

3) The results based off of the RNAfold predictions (e.g. lines 361-363 and 448-455) are not robust enough for making conclusions about potential structural differences between viruses. Minimum free energy predictions can be extremely volatile and may vary wildly based on the sequence length selected for inclusion. Importantly, they are also sensitive to even small sequence variations. For RNAs < ~700 nt RNAfold is expected to predict ~70% of base pairs correctly for an average RNA – but this value can vary greatly depending on the target. If the authors want to perform a rigorous analysis of RNA structure they can attempt to use the available biochemical and in silico structural results for the S region to build a model for this domain of SARS-CoV-2 and attempt to predict the effect of variations in sequence with respect to this model. While interesting, however, I would recommend potentially removing these analyses from the Results section but to discuss the potential roles of RNA structure in these processes in the Discussion, which is already done. I’d not delay the publication of this paper for these analyses, but perhaps include them in a follow-up publication.

Minor Points:

Line 107: This sentence would flow better if "Despite" was replaced with "Although”.

---

## Round 0.2 · accepted · Accept

Thanks for raising the important topic. The reviewers have no more remarks. So, we are happy accept this work for publication.
However, I suggest check again the references and add recent citations in the final version. It is just a suggestion. The data and publications on coronavirus related research appear every day.

·

Basic reporting

The manuscript has been substantially modified according to the reviewers' suggestions. I think this manuscript can be accepted in the current form.

Experimental design

Looks great now. I have no additional comment.

Validity of the findings

Again looks fine

Additional comments

Nicely written manuscript. Detailed and great to read. I wish you all the best for your future endeavours.

Reviewer 2 ·

Basic reporting

no comment

Experimental design

no comment

Validity of the findings

no comment

Additional comments

The authors have done a great job here and I think the revised manuscript is good to go!